# Weaker Effects of the Fourth Dose of BNT162b2 SARS-CoV-2 Vaccine on the Elderly Human Population

**DOI:** 10.3390/vaccines11061095

**Published:** 2023-06-12

**Authors:** Chloé Dimeglio, Isabelle Da-Silva, Marion Porcheron, Marie-Pierre Panero, Laetitia Staes, Pauline Trémeaux, Hélène Villars, Jacques Izopet

**Affiliations:** 1Virology Laboratory, CHU Toulouse, Hôpital Purpan, 31300 Toulouse, Franceporcheron.m@chu-toulouse.fr (M.P.);; 2Inserm UMR 1291—CNRS UMR 5051, Toulouse Institute for Infectious and Inflammatory Diseases (INFINITy), 31300 Toulouse, France; 3Medical School, Toulouse III University, 31000 Toulouse, France; 4Inserm UMR 1295: Center for Research in Population Health (CERPOP), Department of Epidemiology and Public Health, Toulouse, University of Toulouse III, 31073 Toulouse, France; 5Geriatric Department, CHU Toulouse, Toulouse University Hospital, Hôpital Purpan Pavillon Leriche, 31300 Toulouse, France

**Keywords:** vaccination, vulnerable population, SARS-CoV-2, immune monitoring

## Abstract

The vaccines presently available are less effective in older people due to senescence of their immune systems. We measured the antibody responses of 42 adults living in nursing homes after the third and the fourth doses of an mRNA vaccine and found that the strain (BA.2 and BA.2.75: from 64 to 128, BA.5: from 16 to 32, BQ.1.1: from 16 to 64 among the uninfected) influenced the effect of the fourth dose of vaccine on neutralizing antibodies. The fourth dose also increased binding antibodies (from 1036 BAU/mL to 5371 BAU/mL among the uninfected, from 3700 BAU/mL to 6773 BAU/mL among the BA.5 infected). This effect was less significant than that of the third dose of vaccine for both neutralizing (BA.2: from 8 to 128, BA.5: from 2 to 16, BA.2.75: from 8 to 64, BQ.1.1: from 2 to 16) and binding antibodies (from 139.8 BAU/mL to 2293 BAU/mL). However, the fourth dose attained the 5000 BAU/mL threshold conferring approximately 80% protection against a SARS-CoV-2 BA.2 infection in most individuals, unlike the third.

## 1. Introduction

The disease (COVID-19) caused by SARS-CoV-2 virus is particularly severe in older individuals because of their impaired immune response [1]. This is particularly the case with the Omicron variant, for which elderly individuals presented greater disease severity and mortality than the general population [2]. A reduced immunity following vaccination with SARS-CoV-2 messenger RNA vaccines and a rapid decline in the immune response after two vaccine doses have been reported [3]. Some governments have launched a vaccination campaign to protect this population, with a third BNT162b2 dose for nursing home residents (NHRs). A large cohort study among residents of long term care facilities found an association between the third dose of the BNT162b2 vaccine with reduction of SARS-CoV-2 infection, COVID-19 hospitalizations and deaths [4]. The third dose pro-vided 89% protection against infection during the Delta variant surge [5], but a prospective study found that younger participants were more likely to have persistent neutralizing antibodies than older ones after the third dose of vaccine, suggesting that adaptive immunosenescence reduces the booster effect [6]. Another recent study showed the low neutralization antibody level against Omicron after a third dose of vaccine [7], justifying the administration of a fourth dose of vaccination in the elderly to reduce hospitalization and mortality caused by the Omicron variant. A fourth dose of vaccine was therefore recommended for people over 60 in France [8], both to contain the epidemic and provide better protection by increasing SARS-CoV-2 antibody concentrations. An Israeli study reported that a fourth dose of BNT162b2 was associated with lower rates of infection [9], but these studies were all conducted on adults over 60 in such a way that their results cannot be extrapolated to the most vulnerable people.

We have therefore compared the immune responses of nursing home residents (NHRs) to the third and fourth doses of an mRNA vaccine.

## 2. Materials and Methods

### 2.1. Patients

The antibody responses were measured in 43 adults (>65 years) living in 3 French nursing homes. All the individuals were vaccinated with 4 doses of BNT162b. For questions of homogeneity, and to obtain maximum reliability on our results, we chose to exclude from our analysis base the residents who died during the follow-up. Blood samples were taken at the time of their third vaccine dose (28 September 2021–17 November 2021), 3–6 weeks later, at the time of the fourth dose (11 April 2022–16 June 2022), and again 3–6 weeks later. These analyses were performed in the context of the French national SARS-CoV-2 surveillance. According to French law (CSP Art.L1121–1.1) anonymous non-interventional studies do not require institutional review board approval.

### 2.2. Antibodies against SARS-CoV-2 Spike Protein

Samples were analyzed on an Abbott Alinity instrument (Abbott Ireland, Diagnostics Division, Sligo, Ireland). Results above the upper linear limit were diluted and recalculated. Results below the cut-off limit were considered to be negative. The concentrations of binding antibodies were expressed in BAU/mL using the first WHO International Standard for anti-SARS-CoV-2 immunoglobulin (human) as reference for anti-SARS-CoV-2 Ab titers (NIBSC code 20/136; National Institute for Biological Standards and Control, Potters Bar, Hertfordshire, UK). This standard is supplied as a vial containing 250 IU for neutralizing antibody activity equivalent to 250 binding antibody units (BAU) for binding antibody assays. The lyophilisate was suspended in 250 µL ultrapure water and diluted in anti-SARS-CoV-2 Ab-negative serum [10].

### 2.3. Antibodies against SARS-CoV-2 Nucleocapsid Protein

The Abbott SARS-CoV-2 IgG assay detects anti-nucleocapsid antibodies. Samples were analyzed on an Abbott Alinity instrument (Abbott Ireland, Sligo, Ireland). Values of 1.4 and above were considered to be positive.

### 2.4. SARS-CoV-2 Neutralizing Antibodies

Neutralizing antibody titers were determined using a live virus neutralization assay based on Vero cells (ATCC, CCL−81™) and clinical SARS-CoV-2 Omicron BA.2 (GISAID EPI_ISL_), BA.5 (GISAID EPI_ISL_), BA.2.75 (GISAID EPI_ISL_) and BQ1.1 (GISAID EPI_ISL_) strains [10]. Briefly, 104 cells were mixed with the virus suspension (100 50% tissue culture dose [TCID50]) and the tested serum and incubated for 4 days in 96-well plates. Two-fold serial dilutions (from 1:2 to 1:2048) of each serum were tested. The plates were examined to identify the wells showing a cytopathic effect (CPE). The titer was defined as the reciprocal of the highest serum dilution protecting cells from a CPE. The specificity of the neutralization assay evaluated on 100 sera from uninfected individuals was 100%. Its sensitivity was assessed in post-infection settings, showing that 95.3% of sera with SARS-CoV-2 antibodies detected using the Wantaï ELISA contained neutralizing antibodies. Similarly, 99.5% of postvaccination sera with SARS-CoV-2 antibodies detected by ELISA contained neutralizing antibodies [10].

### 2.5. Molecular Diagnosis

SARS-CoV-2 infections were detected using a nucleic-acid amplification method (Aptima^TM^, Hologic, Marlborough, MA, USA). SARS-CoV-2 variant was characterized after RNA extraction from nasopharyngeal swab samples (MGI Easy Nucleic Acid Extraction kit) and long-read sequencing (Pacific Biotechnology, Menlo Park, CA, USA) as previously described [11].

### 2.6. Statistical Analyses

Analyses were performed using Stata v14.0 (StataCorp LLC, College Station, TX, USA). The data were compared using the Wilcoxon signed rank test for matched paired data. Statistically significant difference was defined as a *p*-value < 0.05.

## 3. Results

### 3.1. Patient Characteristics

None of the 43 NHRs (median age: 90 years; range: 62–103; 31 (73.1%) women) had acquired a SARS-CoV-2 infection before their third dose of vaccine (negative anti-*n* antibodies and no positive SARS-CoV-2 RNA). The French government and WHO recommendations regarding the interval between the full regimen and the first booster dose and between the first and the second booster dose are 168 days (6 months). The median time between the second and third vaccine injections was 216 days (IQR 197–232), which was not significantly different from the interval between the third and fourth doses of vaccine (median 252 days, IQR: 196–258, *p.* = 0.31 Wilcoxon signed rank test). About half (23; 54.8%; median age: 91 years; range: 62–100) of the NHRs became infected with BA.5 between the third and the fourth vaccine doses. These infections occurred 153 days (IQR: 124–155) after the third dose of vaccine and 47 days (IQR: 46–77) before the fourth dose.

### 3.2. Effect of the Third and the Fourth Vaccine Dose on NAb

The median neutralizing antibody (NAb) titer of the two-dose vaccinated NHRs was 8 for the BA.2 (IQR: 2–64) and the BA.2.75 (IQR: 4–16) strains at the time of the third injection. It was 2 for both the BA.5 (IQR: 0–16) and the BQ1.1 (IQR: 0–4) strains. 

### 3.3. Effect of the Third Vaccine Dose

The median BA.2 NAb titer increased from 8 (IQR: 2–64) to 128 (IQR: 64–256, *p.* < 0.01 Wilcoxon signed rank test, Figure 1A) one month after the third injection. The median BA.2.75 NAb titer increased from 8 (IQR: 4–16) to 64 (IQR: 32–128, *p* < 0.01 Wilcoxon signed-rank test, Figure 1B) one month after the third injection. The median NAb titer also increased after the third injection for the BA.5 (from 2, IQR: 0–16 to 16, IQR: 4–64, *p* < 0.01 Wilcoxon signed-rank test, Figure 1C) and the BQ1.1 (from 2, IQR: 0–4 to 16, IQR: 8–32, *p* < 0.01 Wilcoxon signed rank test, Figure 1D) strains. 

### 3.4. Effect of the Fourth Vaccine Dose in Uninfected Individuals

The median NAb titer also increased one month after the fourth injection for the BA.2 strain (from 64, IQR: 16–128 to 128, IQR: 64–512, *p* < 0.01 Wilcoxon signed-rank test, Figure 1A) and the BA.2.75 strain (from 64, IQR 64–128 to 128, IQR: 128–256, *p* < 0.01 Wilcoxon signed-rank test, Figure 1C) among those who were not infected. Similarly, the BA.5 and BQ1.1 NAb titers increased one month after the fourth dose: from median 16 (IQR: 4–96) to 32 (IQR: 16–128, *p* < 0.01 Wilcoxon signed-rank test, Figure 1B) for the BA.5 strain, and from 16 (IQR: 8–32) to 64 (IQR: 24–96, *p* < 0.01 Wilcoxon signed-rank test, Figure 1D) for the BQ1.1 strain. 

### 3.5. Effect of the Fourth Dose on Individuals Infected between the Third and Fourth Vaccine Doses

The BA.2 NAb titers of the NHRs who became infected with BA.5 were not different between the third vaccine dose (median 128, IQR: 64–256 one month later) and the fourth dose (median 64, IQR 32–256 at the time of the fourth injection, *p* = 0.67 Wilcoxon signed rank test, Figure 1A). Conversely, the BA.5 NAb titers increased significantly, from 16 (IQR: 4–64) one month after the third dose to 64 (IQR: 32–128) at the time of the fourth injection (*p* < 0.01, Wilcoxon signed rank test, Figure 1B); the median BA.2.75 NAb titers increased from 64 (IQR: 32–128) to 128 (IQR: 128–256, *p* < 0.01 Wilcoxon signed-rank test, Figure 1C) and the median BQ1.1 NAb titers increased from 16 (IQR: 8–32) to 32 (IQR: 16–128, *p* < 0.01, Wilcoxon signed rank test, Figure 1D). 

The fourth dose of vaccine significantly increased the median BA.2 NAb titers (256, IQR: 128–512, *p* < 0.01 Wilcoxon signed rank test, Figure 1A) and the median BQ1.1 titers (128, IQR: 64–256, *p* < 0.01 Wilcoxon signed rank test, Figure 1D) but neither the BA.5 NAb titers (64, IQR: 64–256, *p* = 0.1 Wilcoxon signed rank test, Figure 1B) nor the BA.2.75 NAb titers (192, IQR: 128–512, *p* = 0.2 Wilcoxon signed rank test, Figure 1C). 

### 3.6. Effect of the Third and Fourth Doses on BAb

#### 3.6.1. Third Dose

The median binding antibody (BAb) concentration was 16 times greater one month after the third dose of vaccine (2293 BAU/mL, IQR: 976–3776) than at the time of injection (139.8 BAU/mL, IQR: 38–620.9, *p* < 0.01 Wilcoxon signed rank test, Figure 2).

#### 3.6.2. Fourth Dose in Uninfected Individuals

The median binding antibody (BAb) concentration of the NHRs with no prior SARS-CoV-2 infection one month after the fourth dose (5371 BAU/mL, IQR: 1877–9179) was 5 times greater than the concentration at the time of injection (median 1036 BAU/mL, IQR: 532–2860, *p* < 0.01 Wilcoxon signed rank test, Figure 2).

#### 3.6.3. Fourth Dose on Individuals Infected between the Third and the Fourth Antibody Doses

The BAb concentrations of individuals infected with BA.5 between the third and the fourth doses increased 1.6-fold (from 2293 BAU/mL, IQR: 973–3776 one month after the third injection to 3700 BAU/mL, IQR: 1680–11,712 BAU/mL at the time of the fourth vaccination, *p* < 0.01 Wilcoxon signed rank test, Figure 2). The BAb concentrations were significantly greater (1.8-fold) one month after the fourth dose (6773 BAU/mL, IQR: 4650–14,250, *p* < 0.01 Wilcoxon signed rank test, Figure 2).

## 4. Discussion

Our results suggest that a fourth dose of vaccine increases the immune response to SARS-CoV-2, although its effect on the antibody concentration is weaker than that of the third dose.

The efficacy of a fourth dose in the elderly population was rarely demonstrated. Previous studies estimate that the fourth dose of vaccine achieves more than 70% effectiveness in preventing COVID-19-related deaths in people over the age of 60 [12,13]. However, more than three quarters of the participants were aged under 80, thus limiting the conclusions about the protection offered by the fourth dose for the oldest and most frail people [12,13]. Other studies suggested that a fourth dose prevents premature mortality in the oldest and frailest [14], but antibody concentrations were not measured. We find that the fourth dose of vaccine affects both neutralizing and binding antibodies, although this effect was less significant than that conferred by the third dose of vaccine or a BA.5 natural infection. SARS-CoV-2 BA.5 infection mainly increased the neutralizing antibody response to the BA.5 strain but also significantly increased the neutralizing antibody titers to the BA.2.75 and BQ1 strains, even though this effect remained lower.

The effect of the third dose decreased over time, as found by other recent studies on adults [15,16], but not enough time was spent exploring the decrease in or duration of protection of the fourth dose in residents of long-term care facilities. The decrease in binding antibodies was clear while that of neutralizing antibodies was less pronounced, whatever the strain.

Our findings are consistent with those showing that BA.2.75 is more resistant to neutralization than the ancestral BA.2 strain but has significantly lower neutralization escape than BA.4/5 in both three-dose mRNA-vaccinated individuals and hospitalized Omicron-wave patients [17,18]. Our results are also consistent with those of a recent study showing that serum samples collected following BA.5.1.2 infection could effectively neutralize BA.1, BA.2 and BA.2.75 [19].

The fourth dose of vaccine had a lower relative efficacy than the third dose on the immune response in the subjects who received this last dose very soon after a natural SARS-CoV-2 BA.5 infection, which agrees with the findings of an Israeli study on healthcare workers. They found that a third dose of BNT162b2 vaccine produced a better immunological response than two doses, but the injection of the fourth dose induced a much weaker immunological improvement. This finding joined that of a study carried out in recipients of a fourth dose, which demonstrated that the immunological response following a fourth dose of vaccine after about twenty weeks was equivalent to that of a third dose. [20]. Perhaps the timing of the fourth dose should be designed to maximize its efficiency. For most of our NHRs, the threshold of 5000 BAU/mL conferring approximately 80% protection against SARS-CoV-2 BA.2 [21] was reached with the fourth dose unlike the third, which justified the fourth dose, even in such a short time. BA.5 breakthrough infection has an effect on neutralizing by BQ.1.1 and leading to adapt the booster vaccination schedule in this at-risk population. In a very recent study on a Swedish cohort of vulnerable population, the third dose of mRNA vaccine increased S-binding IgG by 99-fold while the fourth dose of mRNA vaccine induced a relative increase of 3. In this study, the weaker effect of the fourth vaccination, compared to the third, could be explained by the pre-existence of specific antibodies which would limit the reactivation of humoral immunity. The half-life of S-binding IgG was 72 days. As in our study, there was no indication that vaccine-induced antibody levels protected against Omicron infection. In contrast, the size of their cohort allowed them to assess the risk of death which was inversely correlated with IgG levels directed against S below the 20th percentile [22].

Our study has a small number of NHRs, which does not allow us to analyze mortality and questions the generalizability of our results. On the other hand, this small sample was very well monitored (vaccinations and infections), so as to avoid biases that may be present in studies on larger cohorts. These results about mRNA vaccines among the elderly suggest that consideration should be given to the programming of the injection of the fourth dose and future boosts in the most at-risk population, even though an early booster may increase the immune response. It also remains to be seen whether bivalent booster doses will result in longer protection.

## Figures and Tables

**Figure 1 vaccines-11-01095-f001:**
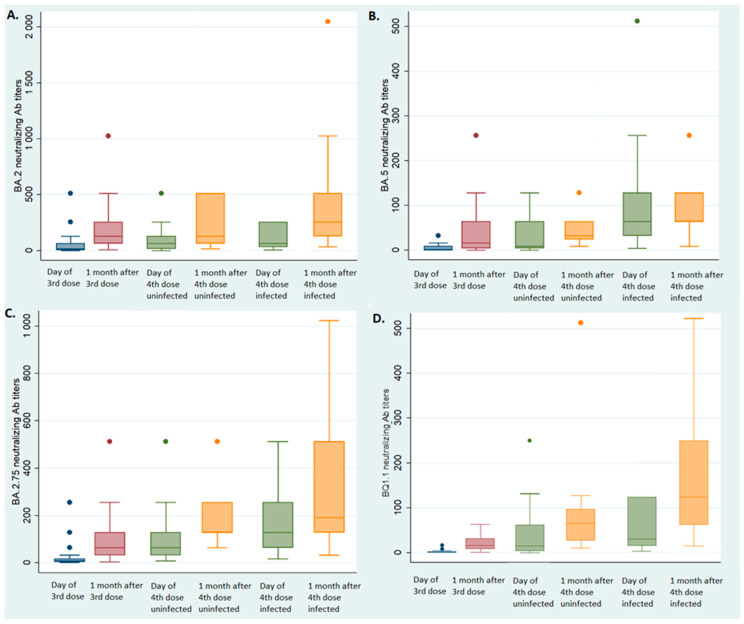
Neutralizing antibody titers according to infected status. Blue box: at the 3rd injection; red box: one month after the 3rd injection; green box: at the 4th injection; yellow box: one month after the 4th injection. (**A**) BA.2 neutralizing antibody titers according to infected status; (**B**) BA.5 neutralizing antibody titers according to infected status; (**C**) BA.2.75 neutralizing antibody titers according to infected status; (**D**) BQ1.1 neutralizing antibody titers according to infected status.

**Figure 2 vaccines-11-01095-f002:**
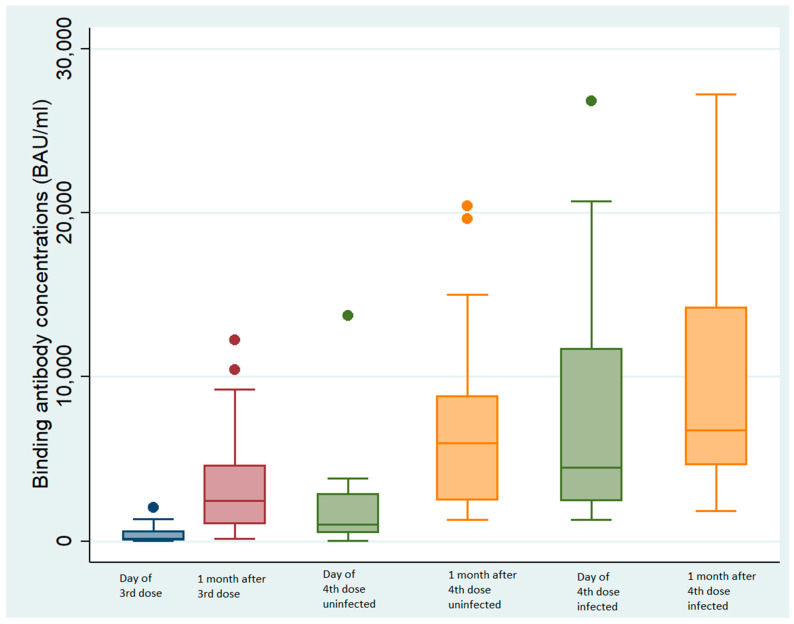
Binding antibody concentrations according to infected status. Blue box: at the 3rd injection; red box: one month after the 3rd injection; green box: at the 4th injection; yellow box: one month after the 4th injection.

## Data Availability

The data presented in this study are available on request from the corresponding author.

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
