# Peer review of "Weaker Effects of the Fourth Dose of BNT162b2 SARS-CoV-2 Vaccine on the Elderly Human Population"

_vaccines, 2023, doi:10.3390/vaccines11061095_

Round 1

Reviewer 1 Report

Dear Author,

Greetings!

1 Effects of the fourth dose of BNT162b2 SARS-CoV-2 vaccine on the elderly ( please revise the title instead of elderly we may use aged human population or senior citizens etc and also what effects please mention clearly in the title)

2A large cohort study among residents of long term care facilities found an association between the third dose of the BNT162b2 vaccine with reduction of SARS-CoV-2 infection, COVID-19 hospitalizations and deaths [3] The third dose provided 89% protection against infection during the Delta variant surge [4] but a recent prospective study found that younger participants were more likely to have per-sistent neutralizing antibodies than older ones after the third dose of vaccine, suggesting that adaptive immunosenescence reduces the booster effect [5].(Please revise this based on updated data)

3Antibodies against SARS-CoV-2 nucleocapsid protein

The Abbott SARS-CoV-2 IgG assay detects anti-nucleocapsid antibodies. Samples

were analyzed on an Abbott Alinity instrument (Abbott Ireland, Sligo, Ireland). Values of

1.4 and above were considered to be positive.

SARS-CoV-2 neutralizing antibodies

Neutralizing antibody titers were determined using a live virus neutralization assay

based on Vero cells (ATCC, CCL-81™) and clinical SARS-CoV-2 Omicron BA.2 (GISAID

EPI_ISL_ 13540703), BA.5 (GISAID EPI_ISL_14238152), BA.2.75 (GISAID

EPI_ISL_16984909) and BQ1.1 (GISAID EPI_ISL_15614157) strains [8].(These serodiagnostic tests based on antigen antibody neutralisation should be explained clearly both acute and chronic titers)

4None of the 43 NHRs (median age: 90 years; range: 62-103), 31 (73.1%) women) had

acquired a SARS-CoV-2 infection before their third dose of vaccine (negative anti-N antibodies

and no positive SARS-CoV-2 RNA). The median time between the second and third

vaccine injections was 216 days (IQR 197-232), which was not significantly different from

the interval between the third and fourth doses of vaccine (median 252 days, IQR: 196-258,

p=0.31 Wilcoxon signed rank test). About half (23; 54.8%) of the NHRs became infected

with BA.5 between the third and the fourth vaccine doses. These infections occurred 153

days (IQR: 124-155) after the third dose of vaccine and 47 days (IQR: 46-77) before thefourth dose.(please specify age based on human population group)

5The fourth dose of vaccine had a lower relative efficacy than the third dose on the immune response in the subjects who received this last dose very soon after a natural SARS-CoV-2 BA.5 infection, which agrees with the findings of an Israeli study on healthcare workers. They found that a third dose of BNT162b2 vaccine produced a better immunological response than two doses, but the injection of the fourth dose induced a much weaker immunological improvement. This finding joined that of a study carried out in recipients of a fourth dose, which demonstrated that the immunological response fol-lowing a fourth dose of vaccine after about twenty weeks was equivalent to that of a third dose. [18]. Perhaps the timing of the fourth dose should be designed to maximize its effi-ciency. For most of our NHRs, the threshold of 5,000 BAU/mL conferring approximately 80% protection against SARS-CoV-2 BA.2 [19] was reached with the fourth dose unlike the third, which justified the fourth dose, even in such a short time. BA.5 breakthrough infection has an effect on neutralizing by BQ.1.1, and leading to adapt the booster vaccina-tion schedule in this at-risk population.(Please revise the paragraph with scientific data and results )

6 Dear authors please align all the references as per journal format and plagiarism check specially in introduction part

7references arrangement should as per journal format page no and vol issue etc (use zotero or mendeley as per journal guidelines

8 Dear scientists please check all the vaccines ,attenuated ,live ,dead  etc clealy mention in conclusion 

Minor editing of English language required

Author Response

1 Effects of the fourth dose of BNT162b2 SARS-CoV-2 vaccine on the elderly ( please revise the title instead of elderly we may use aged human population or senior citizens etc and also what effects please mention clearly in the title)

We have followed your recommendations and we have revised the title as following : Weaker effects of the fourth dose of BNT162b2 SARS-CoV-2 vaccine on the aged human population

2          A large cohort study among residents of long term care facilities found an association between the third dose of the BNT162b2 vaccine with reduction of SARS-CoV-2 infection, COVID-19 hospitalizations and deaths [3] The third dose provided 89% protection against infection during the Delta variant surge [4] but a recent prospective study found that younger participants were more likely to have per-sistent neutralizing antibodies than older ones after the third dose of vaccine, suggesting that adaptive immunosenescence reduces the booster effect [5].(Please revise this based on updated data)

We have modified the introduction section based on very recent studies.

3          Antibodies against SARS-CoV-2 nucleocapsid protein

The Abbott SARS-CoV-2 IgG assay detects anti-nucleocapsid antibodies. Samples

were analyzed on an Abbott Alinity instrument (Abbott Ireland, Sligo, Ireland). Values of

1.4 and above were considered to be positive.

SARS-CoV-2 neutralizing antibodies

Neutralizing antibody titers were determined using a live virus neutralization assay

based on Vero cells (ATCC, CCL-81™) and clinical SARS-CoV-2 Omicron BA.2 (GISAID

EPI_ISL_ 13540703), BA.5 (GISAID EPI_ISL_14238152), BA.2.75 (GISAID

EPI_ISL_16984909) and BQ1.1 (GISAID EPI_ISL_15614157) strains [8].(These serodiagnostic tests based on antigen antibody neutralisation should be explained clearly both acute and chronic titers)

We have completed the two paragraphs concerning serology in order to give more details.

4          None of the 43 NHRs (median age: 90 years; range: 62-103), 31 (73.1%) women) had

acquired a SARS-CoV-2 infection before their third dose of vaccine (negative anti-N antibodies

and no positive SARS-CoV-2 RNA). The median time between the second and third

vaccine injections was 216 days (IQR 197-232), which was not significantly different from

the interval between the third and fourth doses of vaccine (median 252 days, IQR: 196-258,

p=0.31 Wilcoxon signed rank test). About half (23; 54.8%) of the NHRs became infected

with BA.5 between the third and the fourth vaccine doses. These infections occurred 153

days (IQR: 124-155) after the third dose of vaccine and 47 days (IQR: 46-77) before thefourth dose.(please specify age based on human population group)

We have followed your recommandations and we have added informations about the age of the infected ones. There was no statistical difference with age of those who have not been infected.

5          The fourth dose of vaccine had a lower relative efficacy than the third dose on the immune response in the subjects who received this last dose very soon after a natural SARS-CoV-2 BA.5 infection, which agrees with the findings of an Israeli study on healthcare workers. They found that a third dose of BNT162b2 vaccine produced a better immunological response than two doses, but the injection of the fourth dose induced a much weaker immunological improvement. This finding joined that of a study carried out in recipients of a fourth dose, which demonstrated that the immunological response fol-lowing a fourth dose of vaccine after about twenty weeks was equivalent to that of a third dose. [18]. Perhaps the timing of the fourth dose should be designed to maximize its effi-ciency. For most of our NHRs, the threshold of 5,000 BAU/mL conferring approximately 80% protection against SARS-CoV-2 BA.2 [19] was reached with the fourth dose unlike the third, which justified the fourth dose, even in such a short time. BA.5 breakthrough infection has an effect on neutralizing by BQ.1.1, and leading to adapt the booster vaccina-tion schedule in this at-risk population.(Please revise the paragraph with scientific data and results )

 We have added a discussion point on a very recently published Swedish study. All discussion items described here have been published based on peer-reviewed scientific findings. We don't understand the "revise with scientific data and results"

6          Dear authors please align all the references as per journal format and plagiarism check specially in introduction part

Plagarism point has been adressed previously. References have been checked.

7          references arrangement should as per journal format page no and vol issue etc (use zotero or mendeley as per journal guidelines

It is done

8          Dear scientists please check all the vaccines ,attenuated ,live ,dead  etc clealy mention in conclusion 

Our study concerns only mRNA vaccines. We made it clear in the conclusion.

Reviewer 2 Report

This is a well researched and well presented study of a small sample of vulnerable elderly.

Results

The median time between the second and third vaccine injections was 216 days (IQR 197-232), which was not significantly different from the interval between the third and fourth doses of vaccine (median 252 days, IQR: 196-258, p=0.31 Wilcoxon signed rank test). About half (23; 54.8%) of the NHRs became infected with BA.5 between the third and the fourth vaccine doses. These infections occurred 153 days (IQR: 124-155) after the third dose of vaccine a

[please comment on the length of these interdose intervals in terms of the WHO or French official protocol]

Effect of the third and the fourth vaccine dose on NAb The median neutralizing antibody (NAb) titer of the 2-dose vaccinated NHRs was 8 for the BA.2 (IQR: 2-64) and the BA.2.75 (IQR: 4-16) strains at the time of the third injection. It was 2 for both the BA.5 (IQR: 0-16) and the BQ1.1 (IQR: 0-4) strains. Effect of the third vaccine dose The median BA.2 NAb titer increased from 8 (IQR: 2-64) to 128 (IQR: 64-256, p<0.01 Wilcoxon signed rank test, Figure 1A) one month after the third injection. The median BA.2.75 NAb titer increased from 8 (IQR: 4-16) to 64 (IQR: 32-128 p><0.01 Wilcoxon signedrank test, Figure 1B) one month after the third injection. The median NAb titer also increased after the third injection for the BA.5 (from 2, IQR: 0-16 to 16, IQR: 4-64; p><0.01 Wilcoxon signed-rank test, Figure 1C) and the BQ1.1 (from 2, IQR: 0-4 to 16, IQR: 8-32, p><0.01 Wilcoxon signed rank test, Figure 1D) strains.>”

[One issue is survival (which could be affected by many causes but we use it as an outcome measure anyway). Another is antibody levels. Please comment as you report your results in the main text on the antibody levels in relationship to the criteria of the WHO or key French or other authorities]

This will provide helpful information for other researchers]

Generalisability; What is the generalisability of your results based on this sample size? You comment on the small numbers of very elderly]

Author Response

This is a well researched and well presented study of a small sample of vulnerable elderly.

Results

“The median time between the second and third vaccine injections was 216 days (IQR 197-232), which was not significantly different from the interval between the third and fourth doses of vaccine (median 252 days, IQR: 196-258, p=0.31 Wilcoxon signed rank test). About half (23; 54.8%) of the NHRs became infected with BA.5 between the third and the fourth vaccine doses. These infections occurred 153 days (IQR: 124-155) after the third dose of vaccine a”

[please comment on the length of these interdose intervals in terms of the WHO or French official protocol]

  • French and WHO recommendations regarding the interval between the full regimen and the first booster dose and between the first and the second booster dose are 168 days (6 months). In our case, the administration times for the two boosts are longer than those recommended. We have added this comment.

“Effect of the third and the fourth vaccine dose on NAb The median neutralizing antibody (NAb) titer of the 2-dose vaccinated NHRs was 8 for the BA.2 (IQR: 2-64) and the BA.2.75 (IQR: 4-16) strains at the time of the third injection. It was 2 for both the BA.5 (IQR: 0-16) and the BQ1.1 (IQR: 0-4) strains. Effect of the third vaccine dose The median BA.2 NAb titer increased from 8 (IQR: 2-64) to 128 (IQR: 64-256, p<0.01 Wilcoxon signed rank test, Figure 1A) one month after the third injection. The median BA.2.75 NAb titer increased from 8 (IQR: 4-16) to 64 (IQR: 32-128 p><0.01 Wilcoxon signedrank test, Figure 1B) one month after the third injection. The median NAb titer also increased after the third injection for the BA.5 (from 2, IQR: 0-16 to 16, IQR: 4-64; p><0.01 Wilcoxon signed-rank test, Figure 1C) and the BQ1.1 (from 2, IQR: 0-4 to 16, IQR: 8-32, p><0.01 Wilcoxon signed rank test, Figure 1D) strains.>”

[One issue is survival (which could be affected by many causes but we use it as an outcome measure anyway). Another is antibody levels. Please comment as you report your results in the main text on the antibody levels in relationship to the criteria of the WHO or key French or other authorities]

  • For questions of homogeneity, and to obtain maximum reliability on our results, we chose to exclude from our analysis base the residents who died during the follow-up. As you rightly point out, a death can have multiple causes and we have chosen to avoid introducing potential biases into our study. We have added this information.

This will provide helpful information for other researchers]

Generalisability; What is the generalisability of your results based on this sample size? You comment on the small numbers of very elderly]

  • In any statistical study, the sources of variability can be diverse and numerous. Obviously ours is no exception and the small sample size is one of them. On the other hand, this small sample was very well monitored (vaccinations and infections among others), so as to avoid biases that may be present in studies on larger cohorts. We have discussed this point.